# Formulation and Preparation of Water-In-Oil-In-Water Emulsions Loaded with a Phenolic-Rich Inner Aqueous Phase by Application of High Energy Emulsification Methods

**DOI:** 10.3390/foods9101411

**Published:** 2020-10-05

**Authors:** Seyed Mehdi Niknam, Isabel Escudero, José M. Benito

**Affiliations:** Department of Biotechnology and Food Science, University of Burgos, Plaza Misael Bañuelos s/n, 09001 Burgos, Spain; snx1002@alu.ubu.es (S.M.N.); iescuder@ubu.es (I.E.)

**Keywords:** water-in-oil-in-water (W/O/W) emulsion, response surface methodology (RSM), microfluidization, ultrasonic homogenization, rotor-stator mixing, stability analysis

## Abstract

Currently, industry is requesting proven techniques that allow the use of encapsulated polyphenols, rather than free molecules, to improve their stability and bioavailability. Response surface methodology (RSM) was applied in this work to determine the optimal composition and operating conditions for preparation of water-in-oil-in-water (W/O/W) emulsions loaded with phenolic rich inner aqueous phase from olive mill wastewater. A rotor-stator mixer, an ultrasonic homogenizer and a microfluidizer processor were tested in this study as high-energy emulsification methods. Optimum results were obtained by means of microfluidizer with 148 MPa and seven cycles input levels yielding droplets of 105.3 ± 3.2 nm in average size and 0.233 ± 0.020 of polydispersity index. ζ-potential, chemical and physical stability of the optimal W/O/W emulsion were also evaluated after storage. No droplet size growth or changes in stability and ζ-potential were observed. Furthermore, a satisfactory level of phenolics retention (68.6%) and antioxidant activity (89.5%) after 35 days of storage at room temperature makes it suitable for application in the food industry.

## 1. Introduction

Emulsions are generally used for the encapsulation of bioactive compounds in aqueous solutions. They consist of at least two immiscible liquids (oil and water), one of them dispersed as small droplets in the other [1,2]. Typically, droplet diameters in food systems range from 0.1 to 100 μm [3]. Emulsions can be classified as oil-in-water (O/W) or water-in-oil (W/O) emulsions, depending on whether the dispersed phase is oil or water, respectively. Furthermore, there are several types of multiple emulsions, such as oil-in-water-in-oil (O/W/O) or water-in-oil-in-water (W/O/W) emulsions [4,5]. Emulsifiers are commonly added as stabilizers to obtain a kinetically stable system [6].

The formation, stability, and properties of emulsions depend on the characteristics (polarity, water solubility, viscosity, density, etc.) of the oil phase and the type and concentration of components present in the aqueous phase [2,7,8,9,10,11]. The choice of surfactant used as emulsifier is very important in the food industry because it must be able not only to create and stabilize the dispersed phase droplets, but also be biodegradable and nontoxic [12].

Several studies on encapsulation and delivery of polyphenols have been published in recent years [6,13,14,15,16,17,18,19,20,21,22,23,24,25,26,27,28]. The use of encapsulated polyphenols present in phenolic-rich extracts (e.g., green tea, mango peel, olive leaf or grape seed [27,28,29,30,31]) instead of free molecules improves both the stability and bioavailability of the molecules in vitro and in vivo [32,33] using single and multiple emulsions and nanoemulsions.

Emulsion preparation always involves the use of primary homogenization (direct preparation from two separate liquids) and/or secondary homogenization (droplet size reduction in existing emulsions) [33]. The control of homogenization conditions (temperature, pressure and cycles) is required to obtain emulsions with the desired properties (droplet size, stability, and encapsulation and delivery of biocompounds) [34].

Water-in-oil-in-water (W/O/W) double emulsions are formed by small water droplets within larger oil droplets dispersed in an aqueous continuous phase [2]. They are much better encapsulation systems for hydrophilic polyphenols than O/W emulsions, because the release of polyphenols can be prolonged and better controlled [4,33,35,36,37]. However, both O/W and W/O/W emulsions are highly susceptible to instability, mainly by flocculation, coalescence and Ostwald ripening, that will affect the delivery of encapsulated polyphenols. Reducing the droplet size greatly improves the stability and shelf life of emulsions [38,39], but it should be kept in mind that the very small size and therefore the very large specific surface area of the droplets in nanoemulsions may promote the chemical degradation of encapsulated compounds [40].

In this study, response surface methodology (RSM) was applied to determine the optimal composition and operating conditions for the preparation of W/O/W double emulsions loaded with phenolic rich inner aqueous phase from olive mill wastewater. The emulsion formulation was first optimized and then operating conditions to obtain a double nanoemulsion using high energy emulsification methods were examined. In addition to droplet size analysis, physical and chemical stability of double emulsions over time for the optimal formulation were also evaluated.

## 2. Materials and Methods 

### 2.1. Materials

Olive mill wastewater (OMW) used in this study was obtained by means of a three-phase olive oil extraction and centrifugation system and was kindly provided by Mamalan Agro Industrial Company (Zanjan, Iran). Miglyol 812 oil, a mixture of C6-C12 medium chain triglycerides (MCT), was supplied by Sasol GmbH (Hamburg, Germany). Folin-Ciocalteu reagent and hydrochloric acid (37%) were purchased from VWR International Eurolab (Llinars del Vallès, Spain) and Acros Organics (Geel, Belgium), respectively. Sodium carbonate, methanol, gallic acid, 2,2-diphenyl-1-picrylhydrazyl (DPPH), 6-hydroxy-2,5,7,8-tetramethylchroman-2-carboxylic acid (Trolox), sorbitan monooleate (Span 80) and polyoxyethylene (20) sorbitan monooleate (Tween 80) were purchased from Sigma-Aldrich (Darmstadt, Germany). Milli-Q water (Millipore, St. Louis, MO, USA) was used in all samples.

### 2.2. Preparation of Phenolic Rich Olive Mill Aqueous Phase

OMW was first centrifuged (Eppendorf 5804, Hamburg, Germany) for 30 min at 4000 rpm in order to separate the remaining solid particles, obtaining a liquid phase with a pH value of 4.85 (GLP-21 pH-meter, Crison, Barcelona, Spain). Afterwards, based on the method described by Bazzarelli et al. [41], acidification with HCl (37%) was performed by 0.003% (*v*/*w*) addition to reach pH = 1.8. After 24 h, the pretreated OMW was used in a two-stage membrane treatment process including ultrafiltration (UF) followed by nanofiltration (NF).

Both UF and NF were performed in batch concentration mode using a stainless steel HP4750 high-pressure stirred cell of 300 mL capacity supplied by Sterlitech Corporation (Kent, WA, USA). For this purpose, an UF flat sheet polysulfone membrane (US100, 100 kDa, Microdyn-Nadir, Wiesbaden, Germany) was used and the permeate obtained from UF was entered to the same stirred cell module equipped with a NF flat sheet polyamide-thin film composite membrane (NF90, 200 Da, Dow Filmtec, Minneapolis, MN, USA). Transmembrane pressures of 5 bar and 10 bar for UF and NF processes, respectively, were supplied by a nitrogen cylinder. Membrane surface area was 14.6 cm^2^ and both UF and NF treatments were performed at room temperature. The retentate solution obtained by NF treatment was used as the inner aqueous phase in the optimal formulation of W/O/W nanoemulsions.

### 2.3. Preparation of Primary Emulsion (W/O)

W/O emulsions were prepared by mixing Miglyol 812 and Span 80 in amounts described by response surface methodology (RSM) and shown in Table 1. Then the aqueous dispersed phase was added dropwise and the mixture was stirred for 10 min at 500 rpm. The prepared W/O emulsions were immediately used as the dispersed phase in the preparation of W/O/W double emulsions.

### 2.4. Preparation of Double Emulsion (W/O/W)

W/O/W emulsions were prepared by the dropwise addition of the dispersed phase (W/O emulsion) to a continuous phase formed by Milli-Q water and the Tween 80 surfactant.

The primary goal for preparation of double emulsion was optimization of formulation. For this purpose, emulsification process was performed with 26 different compositions based on RSM experimental design (Table 1). The emulsification was performed by using a high intensity ultrasonic homogenizer, described below, for 6 min effective time, in 5 s pulses (5 s off and 5 s on) and 50% amplitude.

The second goal was emulsification of optimized formulation with different high energy methods in order to achieve the optimal operating conditions. A high speed blender (Miccra D9 equipped with a DS-5/K-1 rotor-stator, ART Prozess & Labortechnik, Mülheim, Germany), a high intensity ultrasonic homogenizer (Sonics VCX 500, 500 W, 20 kHz, Newtown, CT, USA) with a titanium alloy microtip probe of 3 mm diameter, and a microfluidizer high shear fluid processor (LM20, Microfluidics, Westwood, MA, USA) were the equipment used for the formation of W/O/W emulsions.

### 2.5. Droplet Size Analysis of the Emulsions

Droplet size distribution, mean droplet diameter and polydispersity index (PDI) of samples were measured by dynamic light scattering (DLS) using a Zetasizer Nano ZS apparatus (Malvern Instruments Ltd., Malvern, UK). Measurements were performed by triplicate at 25 °C. The PDI is a dimensionless measure of the width of the size distribution ranging from 0 to 1, a higher value being indicative of a broader distribution of particle size.

### 2.6. ζ-Potential

ζ-potential was also measured with the aforementioned Zetasizer Nano ZS apparatus, using the laser Doppler velocimetry technique. The measurement was conducted six times for each diluted sample at 25 °C. The ζ-potential was calculated using Henry’s equation and the Smoluchowski approximation.

### 2.7. Total Phenolic Content

Total phenolic content (TPC) of samples was measured using Folin-Ciocalteu standard method [42] with some modifications. Briefly, a standard calibration curve (*R*^2^ = 0.998) was prepared using gallic acid solution. 100 µL of sample was added to a test tube. Then 2.8 mL of distilled water, 2 mL of 7.5% sodium carbonate solution and 100 μL of Folin-Ciocalteu reagent were added to the test tube and mixed. After 60 min of incubation in darkness at room temperature, the absorbance was measured spectrophotometrically at 750 nm wavelength (U-2000 spectrophotometer, Hitachi, Tokyo, Japan). TPC was expressed as milligrams of gallic acid equivalents per liter of sample (mg GA/L).

### 2.8. Antioxidant Activity

Antioxidant activity (AA) of the samples was measured by the DPPH free radical scavenging activity method described by Shen et al. [43]. Briefly, 1 mg of DPPH reagent (2,2-diphenyl-1-picrylhydrazyl) was dissolved in 50 mL of methanol and stored in darkness at 4 °C for 4 h. In order to perform DPPH assay, 60 μL of sample were mixed with 2940 μL of DPPH reagent solution. For the control sample 60 μL of the solvent (in this case Milli-Q water) were used. The mixtures were allowed to stand in darkness at room temperature for 60 min. Then the absorbance was measured at 517 nm wavelength using the U-2000 spectrophotometer. A standard calibration curve (*R*^2^ = 0.995) was prepared using Trolox solution, and antioxidant activity was expressed as milligrams of Trolox equivalents per liter of sample (mg Trolox/L).

### 2.9. Chemical Stability Measurements

The percentage of phenolic compounds held within the aqueous phase after 35 days of storage at room temperature was measured following the method proposed by Regan and Mulvihill [44]. Thus, 3 g of optimal double emulsions were mixed with 3 g of phosphate buffer solution (pH 7) and centrifuged (Eppendorf 5804 centrifuge) at 4500 rpm for 90 min. Then, the lower phase was collected carefully for total phenolic content (TPC) and antioxidant activity (AA) analysis. The percentage of encapsulated compounds (E) was identified by using Equation (1):(1)$$E\left(\%\right)=\left(1-\frac{C_{2}}{C_{1}}\right)\times 100$$
where C_2_ is the concentration of phenolic compounds found in the aqueous phase after centrifugation and C_1_ is the initial concentration of phenolic compounds in the inner aqueous phase [22,44].

### 2.10. Physical Stability Measurements

Stability of W/O/W emulsions was measured in terms of their droplet growth ratio. Since emulsions tend to aggregate during storage, the droplet size of the emulsions was measured after 1 day and also 35 days after preparation. Two different storage conditions were evaluated: 4 °C and room temperature in darkness. Furthermore, optical characterization of the optimal double emulsion was done by static multiple light scattering (S-MLS) using a Turbiscan Lab Expert equipment (Formulaction Co., L’Union, France). The apparatus send a light beam from an electroluminescent diode (λ = 880 nm) through a cylindrical glass cell containing the sample. The emulsion sample (20 mL) without dilution was placed in a cylindrical glass cell and two synchronous optical sensors received the light transmitted through the sample (180° from the incident light) and the light backscattered by the droplets in the sample (45° from the incident light). The optical reading head scans the height of the sample in the cell (about 40 mm), by acquiring transmission and backscattering data every 40 μm. Transmitted and backscattered light were monitored as a function of time and cell height for 35 days at 25 °C [45,46].

### 2.11. Experimental Design

Data were analyzed by a response surface methodology (RSM) using a central composite design (CCD) of type 2^3^ + star with two replicates of the central point. Statgraphics Centurion 18 software (Statgraphics Technologies, Inc., Warrenton, VA, USA) was used in this study. The effects of four variable factors, each with 3 levels, were studied on the droplet size of the nanoemulsions, this being the response variable (Y). The following factors were studied: aqueous phase content in W/O emulsions (X_1_: 4-20% *w*/*w*), surfactant content in W/O emulsions (X_2_: 4–20% *w*/*w*), W/O content in W/O/W emulsions (X_3_: 10–40% *w*/*w*), and surfactant content in W/O/W emulsions (X_4_: 4–10% *w*/*w*). The model generated 26 experimental runs shown in Table 1. The following second-degree polynomial equation (Equation (2)) was used to express the predicted response (Y) as a function of the independent variables (X_1_, X_2_, X_3_ and X_4_):(2)$$Y=a_{0}+a_{1}X_{1}+a_{2}X_{2}+a_{3}X_{3}+a_{4}X_{4}+a_{11}X_{1}^{2}+a_{12}X_{1}X_{2}+a_{13}X_{1}X_{3}+a_{14}X_{1}X_{4}+a_{22}X_{2}^{2}\;+a_{23}X_{2}X_{3}+a_{24}X_{2}X_{4}+a_{33}X_{3}^{2}+a_{34}X_{3}X_{4}+a_{44}X_{4}^{2}$$
where Y represents the response variable (droplet size in this case), a_0_ is a constant, and a_i_, a_ii_ and a_ij_ are the linear, quadratic and interactive coefficients, respectively.

After optimization of the W/O/W formulation, three experimental designs based on RSM were applied in order to determine the optimal emulsification method. The three aforementioned apparatus were tested: a rotor-stator mixer, an ultrasonic homogenizer and a microfluidizer processor. For this purpose, the CCD model generated 10 experimental runs with two replicates of the central point for each emulsification method, as shown in Table 2. The factors selected were rotation speed (X_1_: 11,000–29,000 rpm) and time (X_2_: 5–15 min) for the rotor-stator mixer, sonication time (X_1_: 5–15 min) and amplitude (X_2_: 20–60%) for ultrasonic homogenizer (while temperature (30 °C) and pulses (5 s off and 5 s on) were kept constant), and pressure (X_1_: 50–150 MPa) and number of cycles (X_2_: 1–11) for microfluidizer processor. The related data can be seen in Table 2. The following polynomial equation (Equation (3)) was used to express predicted responses (Y) as a function of the independent variables (X_1_, X_2_) under study:(3)$$Y=a_{0}+a_{1}X_{1}+a_{2}X_{2}+a_{3}X_{3}+a_{11}X_{1}^{2}+a_{22}X_{2}^{2}+a_{12}X_{1}X_{2}$$
where Y represents the response variable (droplet size), a_0_ is a constant, and a_i_, a_ii_ and a_ij_ are the linear, quadratic and interactive coefficients, respectively.

The value of the factors and their effect on the response was determined by analysis of variance (ANOVA) and LSD (lesser significant difference) test. The model was adjusted by means of multiple linear regressions (MLR) and its validity was determined by ANOVA. The level of significance of each coefficient was evaluated through the values of the statistical parameters *F* and *p* (probability), with a confidence level of 95% [47].

## 3. Results and Discussion

### 3.1. Determination of W/O/W Double Emulsion Formulation

#### 3.1.1. Model Fitting of Formulation

One response variable, the droplet size, and four experimental factors were used on the CCD for the optimization of formulation, as shown in Table 1. PDI of the LSD measurements is also shown in Table 1. The default model is quadratic with 15 coefficients, and it has been fit to the response variable. The R-squared statistic indicates that the model as fitted explains 80.57% of the variability in particle size. Three of the coefficients (a_3_, a_4_, and a_34_) of the quadratic polynomial model, Equation (2), have *p*-values less than 0.05, indicating that they are significantly different from zero at the 95% confidence level (Table 3). F-ratio values indicate that, for the range of studied variables, W/O content in W/O/W composition (X_3_) had stronger influence on the droplet size of the emulsions than the other independent variables. F-ratio values also indicate that the interaction with the highest incidence was the one occurring between the W/O concentration in W/O/W (X_3_) and surfactant content in W/O/W emulsion (X_4_).

#### 3.1.2. Response Surface Analysis

In order to study the effect of the independent variables on the droplet size, surface responses of the quadratic polynomial model were generated by varying two of the independent variables within the experimental range while holding the other two constant at the central points. Figure 1a was generated by varying the surfactant content in W/O (X_2_) and surfactant content in W/O/W (X_4_), keeping constant the aqueous phase content in W/O (X_1_) and W/O content in W/O/W (X_3_) at their central values. It shows that increasing X_4_ in the lower levels of X_2_ causes an increase in emulsion particle size which is unfavorable, while for higher levels of X_2_ no considerable variations in particle size were observed. The effect of X_1_ and X_4_ on the particle size of the emulsion at a fixed content of X_2_ and X_3_ in their central values can be seen in Figure 1b. This figure shows that the increase in X_1_ and X_4_ hardly affects the particle size, except at the higher levels for both factors. The effect of X_1_ and X_2_ changing on particle size at the central values of X_3_ and X_4_ is depicted in Figure 1c. It shows that changes of X_1_ and X_2_ hardly affect the particle size. The effects of X_3_ and X_4_ variations on particle size at central values of X_1_ and X_2_ are shown in Figure 1d: a particle size increase is observed as X_3_ increases, being this effect greater at higher levels of X_4_. Figure 1e depicts the effect of X_2_ and X_3_ on particle size at central values of X_1_ and X_4_. Somehow, X_2_ variation hardly modifies the particle size, while increasing X_3_ has a significant effect on particle size growth. Figure 1f shows the effect of X_1_ and X_3_ variation on particle size at central levels of X_2_ and X_4_. A significant growth in particle size with the increase in X_1_ is only observed at the highest levels of X_3_ factor. It can be concluded that these figures prove the significance of X_3_ and X_4_ over other factors on particle size, as well as the interaction between both factors represented in Figure 1d.

#### 3.1.3. Optimization of Double Emulsion Formulation

Following Stoke’s law, the stability of emulsion would increase as the droplet size decreases. Furthermore, the emulsion with higher resistance and control to creaming should also be homogenously distributed in particle size [48]. Numerical optimization of W/O/W emulsion formulation loaded with phenolic rich inner aqueous phase was carried out through design expert software, using desirability function. The W/O/W optimal formulation is expected to be those leading to a stable emulsion with minimum droplet size. Optimum formulation with 98.8% desirability was predicted for 20% (*w*/*w*) of aqueous phase content (X_1_), 4% (*w*/*w*) Span 80 surfactant content (X_2_), 10% (*w*/*w*) of aqueous phase (W/O) content (X_3_), and 4.6% (*w*/*w*) of Tween 80 surfactant content (X_4_). Because some of these optimal values are in the lower range of those selected in the CCD matrix, data extrapolation was performed. It was carried out through expert design software to expand factor input levels and to evaluate the possibility of achieving a better response beyond the levels considered. Formulation optimization process for main response (minimum particle size) yields 100% desirability for 20.3% (*w*/*w*) of aqueous phase content (X_1_), 3.7% (*w*/*w*) Span 80 surfactant content (X_2_), 9.8% (*w*/*w*) of aqueous phase (W/O) content (X_3_), and 4.1% (*w*/*w*) of Tween 80 surfactant content (X_4_). The predicted optimal response is very close to that obtained without extrapolation, which reveals the validity of the experimental design carried out. Three experimental replicates of the optimal formulation were made (20.3% X_1_, 3.7% X_2_, 9.8% X_3_ and 4.1% X_4_). The average droplet size of these samples was 232.5 ± 2.9 nm and the PDI was 0.274 ± 0.013, indicating a narrow distribution in droplet size. As expected, the results were very close to those obtained in run 16 of Table 1.

### 3.2. Effect of the High Energy Emulsification Method on Double Emulsion Droplet Size

Under suitable homogenization conditions, the final droplet size depends strongly on the characteristics of the oil and emulsifier used [34,49,50,51,52,53,54,55,56,57,58,59,60]. In general, small droplet sizes can be obtained more effectively by low-energy approaches than by high-energy approaches, but the former are more limited in the types of oils and emulsifiers that can be used [11].

Microfluidizers generate intense disruptive forces when two fast-moving emulsion streams impinge upon each other within an interaction chamber, leading to highly efficient droplet disruption [11]. Ultrasonic homogenizers use high-intensity ultrasonic waves to create the intense disruptive forces needed to fracture oil and water phases into very small droplets [53,61,62,63]. Ultrasound requires less energy expenditure than other high-energy methods, but sonicator probe-induced contamination is an important drawback. For scale-up applicability, commercial homogenizers based on sonication have been developed in which nanoemulsion is made to flow through a special column capable of producing ultrasonic waves [64]. Rotor-stator mixers are mechanical homogenizers especially used for high viscosity and high disperse phase volume fraction dispersions, e.g., in pharmaceuticals, cosmetics and food processing [65,66,67].

After obtaining the optimal W/O/W double emulsion formulation, the second aim of this work was to achieve the optimal operating conditions for its preparation. For this purpose, the following methods and factors were studied: ultrasonic homogenizer by varying time and amplitude, rotor-stator mixer by varying time and rotor speed, and microfluidizer processor by varying pressure and cycle numbers.

#### 3.2.1. Model Fitting of Emulsification Method

Three experimental designs based on RSM were prepared by the application of rotor-stator mixer, ultrasonic homogenizer and microfluidizer processor. For this purpose, the CCD model generated 10 experimental runs with two replicates of central point for each emulsification apparatus. The average droplet size and PDI of the experiments corresponding to the CCD design is given in Table 2.

The default model is quadratic and statistical models have been fit to the response variables. The R-squared statistic indicates that the fitted model explains 84.23%, 56.13% and 86.18% of the variability in particle size for rotor-stator mixer, ultrasonic homogenizer and microfluidizer processor, respectively. Regarding the rotor-stator mixer, the F-ratio values indicate that the time factor (X_2_) has a stronger influence on the droplet size of the emulsions than the rotational speed factor (X_1_). In the ultrasonic homogenizer, the F-ratio values indicate that time (X_1_) has a slightly stronger influence on the droplet size of double emulsions than the amplitude (X_2_), whereas for microfluidizer processor pressure (X_1_) has considerable stronger influence on the droplet size of the emulsions than the number of cycles (X_2_), as shown in Table 4.

ANOVA showed the significance of the coefficients of the quadratic polynomial models (Equation (3)). Regarding the high speed mixer and microfluidizer, only one coefficient in each model has a *p*-value less than 0.05, indicating that it is significantly different to zero with 95% confidence level. Moreover, no coefficient has a *p*-value less than 0.05 for the ultrasonic homogenizer model (Table 4).

#### 3.2.2. Response Surface Analysis

In order to study the effect of the emulsification methods and operating conditions on the droplet size, surface response plots of the quadratic polynomial model were generated. Figure 2a shows the effects of time and rotation speed in rotor-stator mixer on the emulsion droplet size. This figure shows that regardless of time level, 20,000–23,000 rpm is the optimal range for rotation speed. However, increasing time causes a continuous decrease in particle size which is favorable.

Figure 2b shows the interaction of time and amplitude on the achieved particle size in ultrasonic homogenizer; as it can be observed, amplitude hardly affects the particle size except for lower time levels. Furthermore, increasing time is significant on reducing particle size [68], as it causes an increase in temperature and cavitation intensity which accelerates the breakdown of droplets [57,69]. Somehow, the minimum particle size would be achieved for high time and lower amplitude levels.

In emulsions prepared by microfluidization the mean droplet diameter decreased with increasing homogenization pressure. It is in accordance with the study performed by Bai et al. [49], in which the mean droplet diameter decreased from around 213 to 150 nm as the homogenization pressure increased from 4 to 14 kbar. The decrease in droplet size with increasing pressure can be attributed to the increase in the magnitude of the disruptive forces generated within the homogenization chamber. Therefore double emulsions containing small droplets with a narrow particle size distribution can be produced by using microfluidizer processor. Figure 2c shows the interaction between pressure and cycles and their effect on particle size in microfluidizer emulsification tests. It can be observed that increasing pressure is significantly effective in decreasing particle size at middle cycle levels. Somehow, the minimum particle size can be achieved at 140–150 MPa pressure levels and 6–7 cycles.

#### 3.2.3. Optimization of W/O/W Emulsion Preparation Conditions

The optimal conditions for the emulsification of the phenolic rich olive mill aqueous phase used in this work would be those leading to a stable double emulsion with minimum droplet size. Numerical optimization was performed through design expert software, using desirability function method. Regarding the W/O/W emulsification by rotor-stator mixer, the combined optimum ingredient levels for average particle size with 100% desirability were predicted to be achieved by emulsification at 20,000 rpm and 10 min. The predicted response at optimal value was 169.4 nm for particle size and polydispersity index (PDI) of 0.546, while observed results were 301.7 ± 12.1 nm and 0.439 ± 0.030 for particle size and PDI, respectively (Table 2). This large difference between predicted and observed results is due to imprecision in the average size for heterogeneous populations, as revealed the high PDI values.

In the case of optimal double emulsion prepared by ultrasonic homogenizer, response of 316.15 nm for particle size and PDI of 0.453 with 100% desirability was predicted while emulsification for 10 min at 20% amplitude was the combined optimum factor levels. The observed results, as shown in Table 2, were 338.5 ± 16.5 nm and 0.468 ± 0.021 for particle size and PDI, respectively. The narrow difference between the predicted and observed responses can verify the reliability of the ultrasonic assisted experimental design.

Best results were obtained with microfluidizer processor (Table 2). Numerical optimization performed by the design expert software predicted the optimal factor levels as 148 MPa of pressure and 7 cycles. The predicted response was 96.03 nm and 0.330 for particle size and PDI, respectively, whereas the observed responses for the mentioned factor levels were 105.3 ± 3.2 nm and 0.233 ± 0.020 for particle size and PDI, respectively. The narrow difference between the predicted and observed responses can also verify the reliability of the microfluidizer experimental design. Therefore, it can be concluded from the observed responses of optimal points from three emulsification methods that microfluidizer processor can be considered as the optimal method to achieve double emulsions with lower particle size and PDI.

### 3.3. Effect of the High Energy Emulsification Method on Double Emulsion Droplet Size

#### 3.3.1. Stability Evaluations During Storage

The stability of the optimized double emulsions during their storage was evaluated by two methods. The first one was the comparative evaluation of droplet size (at room temperature and 4 °C) after 1 day and 35 days from emulsification in order to study the possible droplet growth. The second method was the backscattering (BS) evolution over time using Turbiscan Lab Expert apparatus to detect possible creaming, sedimentation, coalescence, flocculation or Ostwald ripening effect. Sedimentation, creaming and flocculation phenomena are unexpected, while those of coalescence and Ostwald ripening are more likely in the behavior of nanoemulsions [55].

Figure 3 shows the changes in droplet size between 1 day and 35 days of storage at 4 °C and room temperature. As it can be seen from Figure 3a, there is no considerable change in droplet size distribution of rotor-stator optimal W/O/W emulsion during the storage period while the intensity increased regardless of storage temperature condition. According to Figure 3b, which shows the droplet size distribution of emulsion prepared in ultrasonic homogenizer, average droplet size did not vary considerably but PDI was increased during the storage period. It is already mentioned the advantage of optimal W/O/W prepared by microfluidizer processor over other devices due to significantly better average droplet size. Furthermore, it showed the best droplet growth stability, as shown in Figure 3c: neither average droplet size nor PDI was changed after 35 days of storage at both 4 °C and room temperature. ζ-potential was also measured for this sample and the results indicated the stable value of −2.7 mV after 1 day and 35 days of emulsification.

There is a direct relationship between the increased emulsion droplet diameter and creaming, which is the formation of a lipid rich cream layer on the liquid surface as a result of the generation of big lipid droplets from smaller ones due to weak steric repulsions [18,33]. The coalescence process mainly takes place after double emulsion production and is thus stronger influenced by geometrical parameters, like inner and outer droplet sizes and dispersed phase concentrations, than by process parameters [59]. Both phenomena negatively affect emulsion stability, a very important factor determining its shelf life in commercial food and beverage applications [33,70].

BS profiles monitored during 35 days at 25 °C are depicted in Figure 4. As it is shown in Figure 4a,b, creaming instability occurs in emulsification by rotor-stator mixer and ultrasonic homogenizer. In contrast, the optimal W/O/W emulsion prepared by microfluidizer processor, Figure 4c, showed no considerable destabilization during the storage period and the sample was also visually stable, maintaining the same bluish color and semitransparent appearance after 35 days.

The rotor-stator mixer sample happened to exhibit visual creaming instability after a few hours of storage. Nevertheless, double emulsion obtained by using ultrasonic homogenizer showed a slightly larger droplet size but visual instability was not observed after the same storage period. Similar results were obtained by Einhorn-Stoll et al. [71], who observed a rapid destabilization of emulsions prepared by a single step with the Ultra-Turrax homogenizer. However, both chemical analysis and physical appearance indicated slight levels of creaming after few days of storage (Figure 4b). The shifting of the curves in Figure 4b,c could be attributed to the presence of air bubbles in cells or being a little shaken during each analysis.

#### 3.3.2. Retention Properties of Nanoemulsions

As it was aforementioned the OMW used in this study was acidified to pH = 1.8 prior to membrane treatment in order to increase the content of selective bioactive compounds. The effect of pH on physical stability of different emulsions has been previously determined by several authors. Recently, the effect of pH on curcumin emulsions was investigated [72]. The authors found that more than 85% of curcumin was present after one month of storage at 37 °C when acidic conditions were employed. However, emulsions at pH 7.0, 7.4 and 8.0 contained only 62, 60, and 53% of the initial curcumin, respectively, thus demonstrating low stability. The effect of pH on the stability of emulsions stabilized by pectin-zein complexes has also been studied by Juttulapa et al. [73]: they found a greater cross-linking polymer network at pH 4 than pH 7, providing thus a smaller droplet size distribution.

Chemical stability of W/O/W emulsion was evaluated by measurements of retention levels of total phenolic content (TPC) and antioxidant activity (AA) after 35 days of storage at room temperature. TPC and AA measurements were done on the W/O/W double emulsion using the optimal formulation prepared by microfluidizer processor at 148 MPa and 7 cycles. Polyphenols are very sensitive compounds that can be easily degraded over time; therefore, their encapsulation in suitable double emulsions is an effective solution to prevent the degradation of phenolic compounds at a satisfactory level. The phenolic rich inner aqueous phase used in this study (after membrane treatments and before emulsification) had a TPC of 1399.8 ± 17.9 mg GA/L and 286.5 ± 0.3 mg Trolox/L of AA. After 35 days of storage at room temperature, 68.6% and 89.5% retentions of TPC and AA, respectively, were preserved. 

In related studies, Akhtar et al. [74] reported 72% rutin and anthocyanin flavonoids retention after 10 days of storage in a W/O/W nanoemulsion system, but a polymodal droplet size distribution and inner phase leakage were observed. Gomes et al. [75] studied the retention of gallic acid in W/O and O/W emulsion systems with soybean oil, obtaining a 15% reduction in gallic acid content after 7 days of storage. Mohammadi et al. [22] reported 22% of phenolic compounds release by preparing double emulsions stabilized only with whey protein concentrate (WPC) containing olive leaf extract after 20 days. Furthermore, in the study performed by Gadkari et al. [18] it was noted that when the emulsions were stored at temperatures of 277 K, 300 K and 310 K, the green tea polyphenols present in emulsions were degraded by 4.25%, 15.97% and 22.78%, respectively. It can be concluded that the results obtained in the present study are in the usual retention range of phenols encapsulated in emulsions with applications for the food industry.

## 4. Conclusions

Response surface methodology was applied in this work to determine both the optimal composition and operating conditions for preparation of W/O/W nanoemulsion loaded with phenolic rich inner aqueous phase from olive mill wastewater. The optimal formulation for primary W/O emulsion was 20.3% (*w*/*w*) of phenolic rich aqueous solution as dispersed phase and 3.7% (*w*/*w*) of Span 80 in MCT oil as continuous phase. The optimal composition for W/O/W emulsion was 9.8% (*w*/*w*) of W/O as dispersed phase and 4.1% (*w*/*w*) of Tween 80 in Milli-Q water as external phase. Three methods were tested to obtain the optimal emulsification conditions: mechanical homogenization (rotor-stator mixer), ultrasonic homogenization and microfluidization. Optimum results were achieved by microfluidization at 148 MPa and 7 cycles input levels, obtaining a W/O/W nanoemulsion with an average droplet diameter of 105.3 ± 3.2 nm and a polydispersity index of 0.233 ± 0.02. Samples obtained by rotor-stator mixer and ultrasonic homogenizer showed creaming instability after few days of emulsification, while emulsion obtained by microfluidization showed no droplet size growth or changes in stability and ζ-potential after 35 days of storage at 25 °C. Furthermore, it showed a satisfactory level of phenolics retention (68.6%) and antioxidant activity (89.5%) after 35 days of storage, which are suitable for application in the food or pharmaceutical industry.

## Figures and Tables

**Figure 1 foods-09-01411-f001:**
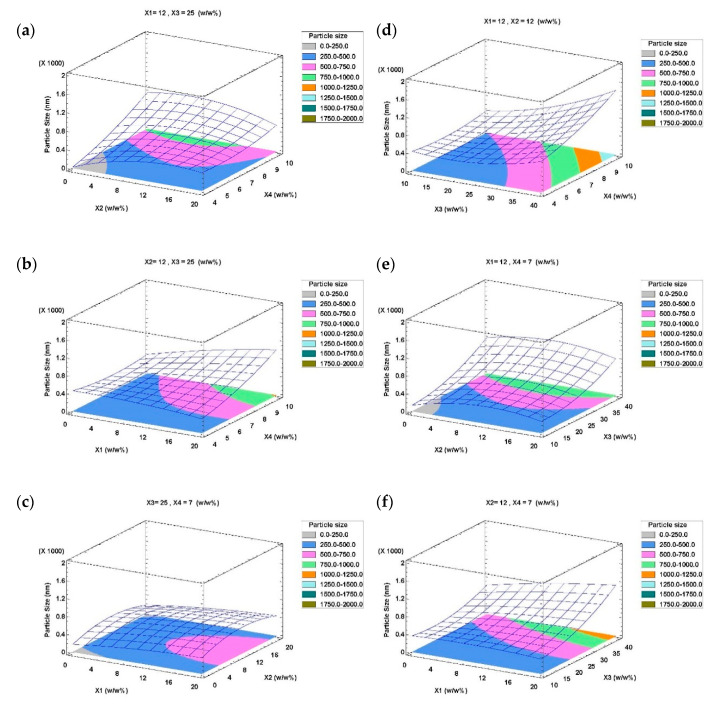
Response surface plots of interactions between two input factors, while holding the other two constants at their central points, on particle size of W/O/W emulsions. X_1_: Aqueous phase content in W/O emulsion; X_2_: Surfactant content in W/O emulsion; X_3_: W/O content in W/O/W emulsion; X_4_: Surfactant content in W/O/W emulsion. (**a**) X_2_ vs. X_4_; (**b**) X_1_ vs. X_4_; (**c**) X_1_ vs. X_2_; (**d**) X_3_ vs. X_4_; (**e**) X_2_ vs. X_3_; (**b**) X_1_ vs. X_3_.

**Figure 2 foods-09-01411-f002:**
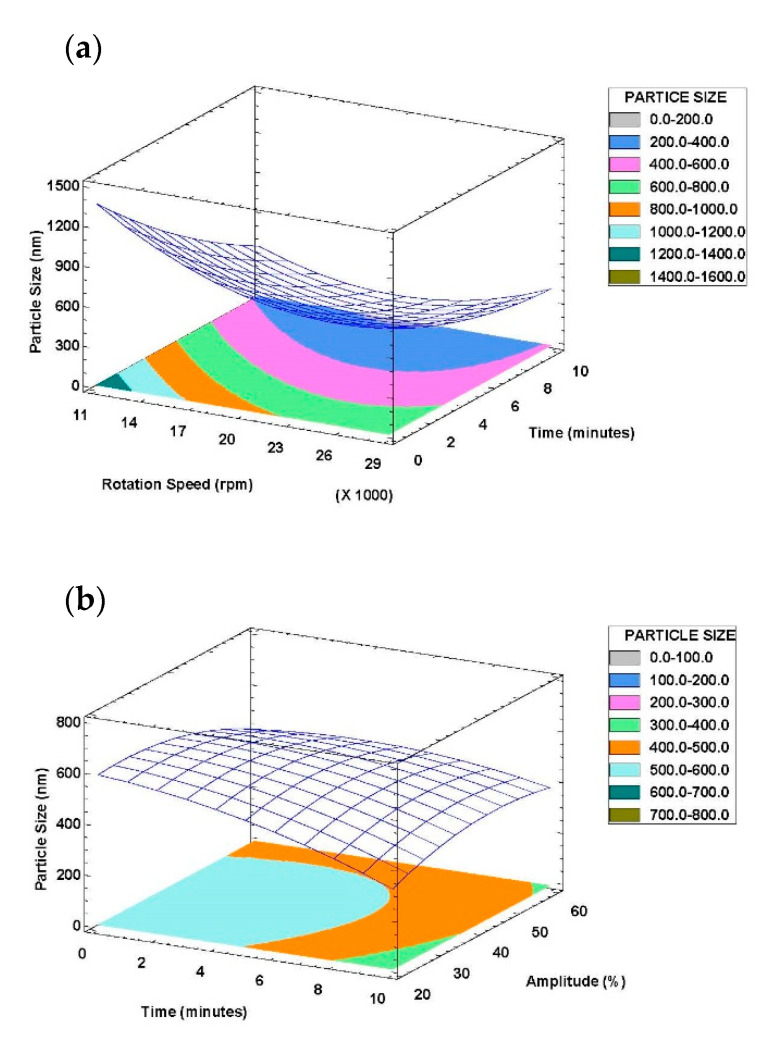
Response surface plots of interaction between emulsification input factors on particle size: (**a**) Rotor-stator mixer; (**b**) Ultrasonic homogenizer; (**c**) Microfluidizer processor.

**Figure 3 foods-09-01411-f003:**
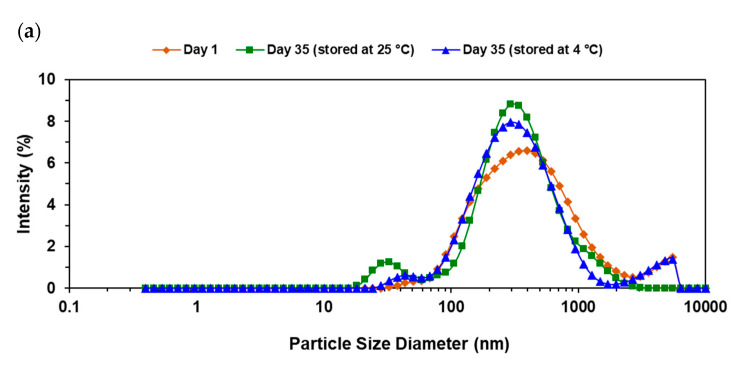
Particle size distribution of W/O/W emulsions (optimal formulations) prepared by three different devices after 1 day, and after 35 days of storage at room temperature and at 4 °C: (**a**) Rotor-stator mixer; (**b**) Ultrasonic homogenizer; (**c**) Microfluidizer processor.

**Figure 4 foods-09-01411-f004:**
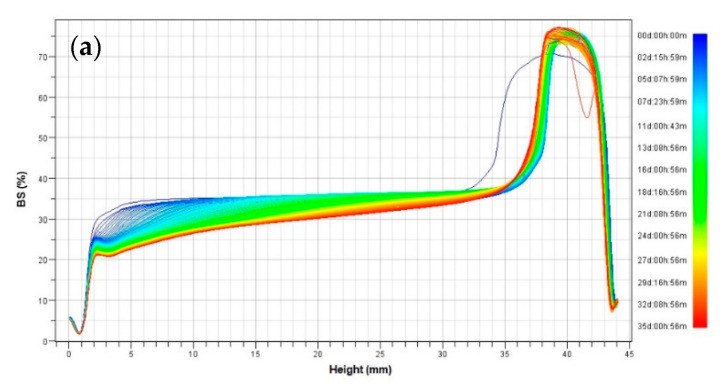
Backscattering profiles of W/O/W emulsions (optimal formulations) prepared by three devices through 35 days of storage at 25 °C: (**a**) Rotor-stator mixer; (**b**) Ultrasonic homogenizer; (**c**) Microfluidizer processor.

**Table 1 foods-09-01411-t001:** Experimental matrix of W/O/W formulations based in a central composite design (CCD).

Run	Independent Variables				Response Variables			
	Aqueous Phase Content in W/O Emulsion (X_1_, % *w*/*w*)	Surfactant Content in W/O Emulsion (X_2_, % *w*/*w*)	W/O Content in W/O/W Emulsion (X_3_, % *w*/*w*)	Surfactant Content in W/O/W Emulsion (X_4_, % *w*/*w*)	Droplet Size (Y, nm)		Polydispersity Index (PDI)	
					Mean	SD	Mean	SD
1	12	12	10	7	470.4	3.7	0.426	0.043
2	20	12	25	7	415.2	12.8	0.357	0.068
3	20	4	40	4	512.2	14.8	0.561	0.033
4	20	20	10	4	279.7	18.1	0.439	0.015
5	12	4	25	7	344.8	10.1	0.486	0.032
6	20	4	40	10	1767	119.5	0.592	0.071
7	4	20	10	4	388.7	6.3	0.458	0.020
8	20	4	10	10	470.3	4.7	0.437	0.024
9	20	20	40	4	478.3	19.1	0.607	0.076
10	12	12	25	4	316.2	13.0	0.571	0.027
11	4	4	10	10	363.5	11.4	0.329	0.054
12	12	20	25	7	371.8	7.4	0.454	0.009
13	12	12	40	7	768.9	17.3	0.620	0.009
14	20	20	40	10	1828	58.3	0.501	0.015
15	12	12	25	7	585.8	19.1	0.553	0.041
16	20	4	10	4	242.6	5.6	0.283	0.011
17	4	20	10	10	330.0	10.4	0.273	0.011
18	4	20	40	4	923.6	24.3	0.610	0.045
19	12	12	25	7	857.4	13.3	0.487	0.047
20	4	4	10	4	232.3	2.5	0.365	0.045
21	20	20	10	10	216.3	5.0	0.460	0.032
22	4	4	40	10	1336	246.6	0.907	0.162
23	12	12	25	10	795.7	11.5	0.514	0.017
24	4	20	40	10	362.5	8.4	0.574	0.018
25	4	12	25	7	476.5	4.2	0.467	0.006
26	4	4	40	4	415.5	37.9	0.872	0.190

W/O: water-in-oil. W/O/W: water-in-oil-in-water. SD: standard deviation.

**Table 2 foods-09-01411-t002:** Experimental matrix based in a central composite design (CCD) for W/O/W double emulsion preparation by three high-energy emulsification methods.

**Run**	**Rotor-Stator Mixer**					
	**Independent Variables**		**Response Variables**			
	**Rotation Speed (X_1_, rpm)**	**Time (X_2_, min)**	**Droplet Size (Y, nm)**		**Polydispersity Index (PDI)**	
			**Mean**	**SD**	**Mean**	**SD**
1	29000	2	735.6	38.1	0.903	0.028
2	20000	2	468.2	4.0	0.585	0.016
3	11000	2	1255	174.6	0.392	0.147
4	11000	6	580.0	21.9	0.487	0.032
5	11000	10	352.9	10.5	0.554	0.076
6	29000	6	530.4	10.2	0.484	0.029
7	29000	10	325.5	2.6	0.539	0.007
8	20000	10	301.7	12.1	0.439	0.030
9	20000	6	474.5	9.7	0.457	0.032
10	20000	6	366.4	9.5	0.551	0.019
**Run**	**Ultrasonic Homogenizer**					
	**Independent Variables**		**Response Variables**			
	**Time (X_1_, min)**	**Amplitude (X_2_, %)**	**Droplet Size (Y, nm)**		**Polydispersity Index (PDI)**	
			**Mean**	**SD**	**Mean**	**SD**
11	10	20	338.5	16.4	0.468	0.021
12	6	60	434.6	16.9	0.436	0.010
13	6	20	388.1	2.4	0.468	0.013
14	2	20	622.7	26.6	0.565	0.101
15	6	40	460.4	3.9	0.432	0.014
16	10	40	394.1	3.1	0.368	0.042
17	10	60	375.2	1.4	0.350	0.043
18	2	60	484.5	3.5	0.429	0.009
19	6	40	667.5	6.8	0.410	0.013
20	2	40	482.3	7.9	0.429	0.004
**Run**	**Microfluidizer Processor**					
	**Independent Variables**		**Response Variables**			
	**Pressure (X_1_, MPa)**	**Cycles (X_2_)**	**Droplet Size (Y, nm)**		**Polydispersity Index (PDI)**	
			**Mean**	**SD**	**Mean**	**SD**
21	50	1	393.5	7.6	0.387	0.037
22	150	11	120.8	1.9	0.316	0.009
23	150	1	128.8	1.9	0.378	0.007
24	100	6	105.8	1.2	0.468	0.009
25	50	6	195.3	6.9	0.281	0.017
26	100	1	253.4	8.2	0.422	0.005
27	100	11	118.5	1.3	0.454	0.004
28	50	11	234.1	6.3	0.424	0.032
29	100	6	187.3	2.8	0.300	0.003
30	150	6	134.8	4.8	0.387	0.053

**Table 3 foods-09-01411-t003:** Analysis of variance of the regression coefficients of the quadratic model (Equation (2)) for the droplet size of W/O/W emulsions.

Source	Regression Coefficients	*F*-Ratio	*p*-Value
a_0_	1029.33	-	-
a_1_	−46.5462	1.27	0.2841
a_2_	75.8451	0.17	0.6882
a_3_	−47.5277	19.37	0.0011
a_4_	−171.888	9.00	0.0121
a_11_	−0.407612	0.02	0.8878
a_12_	0.149121	0.02	0.8973
a_13_	0.861198	2.05	0.1805
a_14_	6.08568	4.08	0.0683
a_22_	−1.77558	0.40	0.5421
a_23_	−0.179323	0.09	0.7714
a_24_	−4.8638	2.61	0.1345
a_33_	0.656502	0.67	0.4309
a_34_	3.78764	5.56	0.0379
a_44_	9.33476	0.22	0.6509

**Table 4 foods-09-01411-t004:** Analysis of variance of the regression coefficients of the quadratic Equation (3) for the W/O/W emulsion preparation by three high-energy emulsification methods.

Source	Rotor-Stator Mixer			Ultrasonic Homogenizer			Microfluidizer Processor		
	Regression Coefficients	*F*-Ratio	*p*-Value	Regression Coefficients	*F*-Ratio	*p*-Value	Regression Coefficients	F-Ratio	*p*-Value
a_0_	2570.49	-	-	497.885	-	-	616.672	-	-
a_1_	−0.139724	2.06	0.2242	−20.3173	3.44	0.1372	−4.52892	13.05	0.0225
a_2_	−166.557	12.69	0.0235	7.39658	0.04	0.8426	−48.2018	6.20	0.0675
a_11_	2.7045 × 10^−6^	3.90	0.1195	−1.80134	0.17	0.6993	0.0107943	0.69	0.4523
a_12_	0.00341667	2.11	0.2203	0.546562	0.68	0.4558	0.1514	2.33	0.2014
a_22_	3.05089	0.19	0.6827	−0.139179	0.64	0.4675	1.91543	2.18	0.2140

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
