# Peer review of "Formulation and Preparation of Water-In-Oil-In-Water Emulsions Loaded with a Phenolic-Rich Inner Aqueous Phase by Application of High Energy Emulsification Methods"

_foods, 2020, doi:10.3390/foods9101411_

Round 1
Reviewer 1 Report
Grammar error:
page 1, line 18: polydispersion index
The paper deals with the formulation and preparation of emulsions. It is completed by coloured graphs illustrating dependencies of the chosen variables.
1.The processed material for experiments was olive mill wastewater. The results are interesting but the microbiological assessment of this water is missing. It is expected to use it for the food or beverage preparation?
2. Great attention in the paper is paid to the size of droplets. The effectivity for the retention of phenolics is devoted just one short chapter (3.3.2). Was the retention of phenolics influenced by the device and its parameters (rotor-stator mixer; ultrasonic homogenizer; microfluidizer processor)?
3. If the measurements has been not done (or has been done in a very limited scope) I recommend to explain the reason and also to shorten the text in the introduction concerning phenolics.
Author Response
Response to Reviewer 1 Comments
Point 1: Grammar error: page 1, line 18: polydispersion index
Response 1: We thank the reviewer for this comment. This grammar error has been corrected in the
abstract (line 18) of the revised version of the manuscript.
Point 2: The paper deals with the formulation and preparation of emulsions. It is completed by
coloured graphs illustrating dependencies of the chosen variables.
1. The processed material for experiments was olive mill wastewater. The results are interesting but
the microbiological assessment of this water is missing. It is expected to use it for the food or
beverage preparation?
Response 2: The main aim of the funded project is to recover the polyphenols present in olive mill
wastewater (OMW) and encapsulate them in stable double emulsions to be used in food and
pharmaceutical applications. A preliminary research to determine the optimal composition and
operating conditions for preparation of stable water-in-oil-in-water (W/O/W) emulsions needed to
encapsulate the polyphenols is presented in this manuscript. The experiments were performed in
Spain using Iranian OMW obtained from of a three phase olive oil extraction and centrifugation
system. We have the physicochemical parameters of the OMW (pH, COD, BOD, etc.), but
unfortunately not the microbiological assessment.
Once the process to obtain stable emulsions has been optimized, the next step will be the use of
these double emulsions with encapsulated polyphenols in several food applications, for which it is
obvious that, as the reviewer has pointed out, we will have to carry out the microbiological analysis
of the olive mill wastewater.
Point 3: 2. Great attention in the paper is paid to the size of droplets. The effectivity for the
retention of phenolics is devoted just one short chapter (3.3.2). Was the retention of phenolics
influenced by the device and its parameters (rotor-stator mixer; ultrasonic homogenizer;
microfluidizer processor)?
3. If the measurements has been not done (or has been done in a very limited scope) I recommend to
explain the reason and also to shorten the text in the introduction concerning phenolics.
Response 3: As it was indicated in the previous response, the main aim of this work is to determine
the optimal composition and operating conditions to obtain stable W/O/W emulsions. Some studies
mentioned in this work indicate that phenolics retention can be affected by the device and operating
parameters used for emulsion preparation, but also that it is more important to obtain a stable
emulsion with 80-85% phenolic retention values that another one with more than 95% phenolics
retention but very low stability. As a general rule, higher phenolics retention can be achieved with
more stable emulsions. This is the reason why we have only carried out the encapsulation of
polyphenols in the double emulsion obtained by microfluidization, which is much more stable than
those obtained by the other two methods. However, we agree with the last comment of the reviewer
and have decided to shorten the text in the introduction concerning phenolics.
Reviewer 2 Report
Overall, it is a well-structured manuscript describing a significant amount of experimental data in the area of nano-emulsions emulsification and droplet characterisation along with interesting analysis and findings of potential use in food and pharmaceutical applications (e.g. nano-encapsulation of polyphenols).
The quality of the final manuscript could possibly merit from consideration of the below specified comments/recommendations (see in italics):
Line-57: “…use of encapsulated polyphenols…” (can you please specify in brackets the most recently studies polyphenols with regards their encapsulation capacities?”)
p-121: “double emulsions…” (should be in plural, “emulsions” instead?)
p-199: Perhaps to specify somewhere in the paragraph that DPPH method measures radical scavenging activities?
p-22-23: “optical characterisation…..was done by S-MLS…”
- (any Reference for the preparation technique? Or protocol set up in the laboratory?)
p-342: “the characteristics of the oil and emulsifier used…” (perhaps to specify which one? e.g. concentration and type?)
References:
- A number of References (about 10%) are old ones of close or older than 20 years! (e.g. Nr 29, 35, 40, 59, 64, 65 etc.). Are they extremely important for the analysis/discussion or perhaps could be replaced by most updated literature evidence?
- In References Nr 29, 67, please highlight date of publication in bold
General comment: It could perhaps enhance quality if you include 1-2 images about the optical characterisation of nanoemulsions e.g. indicating baseline/destabilisation etc
Author Response
Response to Reviewer 2 Comments
Overall, it is a well-structured manuscript describing a significant amount of experimental data in
the area of nano-emulsions emulsification and droplet characterisation along with interesting
analysis and findings of potential use in food and pharmaceutical applications (e.g. nanoencapsulation of polyphenols).
The quality of the final manuscript could possibly merit from consideration of the below specified
comments/recommendations (see in italics):
Point 1: Line-57: “…use of encapsulated polyphenols…” (can you please specify in brackets the
most recently studies polyphenols with regards their encapsulation capacities?”)
Response 1: New recently studies have been added to this paragraph and included in the References
section (Refs. [29–32] in the revised manuscript). The paragraph has been rewritten (lines 40–44)
as: “Several studies on encapsulation and delivery of polyphenols have been published in recent
years [6,13–28]. The use of encapsulated polyphenols present in phenolic-rich extracts (e.g., green
tea, mango peel, olive leaf or grape seed [27–31]) instead of free molecules improves both the
stability and bioavailability of the molecules in vitro and in vivo [32,33] using single and multiple
emulsions and nanoemulsions”.
Point 2: p-121: “double emulsions…” (should be in plural, “emulsions” instead?)
Response 2: Thanks for this comment. The suggested change has been made in the revised version
(page 2, line 60) of the manuscript.
Point 3: p-199: Perhaps to specify somewhere in the paragraph that DPPH method measures radical
scavenging activities?
Response 3: We agree with the reviewer’s indication. The sentence has been rewritten as:
“Antioxidant activity (AA) of the samples was measured by the DPPH free radical scavenging
activity method described by Shen et al. [43].” (page 3, lines 133–134 in the revised manuscript).
Point 4: p-22-23: “optical characterisation…..was done by S-MLS…”
• (any Reference for the preparation technique? Or protocol set up in the laboratory?)
Response 4: The protocol set up in the laboratory was already indicated in Section 2.10: “…optical
characterization of the optimal double emulsion was done by static multiple light scattering (SMLS) using a Turbiscan Lab Expert equipment (Formulaction Co., L’Union, France). The
apparatus send a light beam from an electroluminescent diode (λ = 880 nm) through a cylindrical
glass cell containing the sample. The emulsion sample (20 mL) without dilution was placed in a
cylindrical glass cell and two synchronous optical sensors received the light transmitted through the
sample (180° from the incident light) and the light backscattered by the droplets in the sample (45°
from the incident light). The optical reading head scans the height of the sample in the cell (about
40 mm), by acquiring transmission and backscattering data every 40 µm. Transmitted and
backscattered light were monitored as a function of time and cell height for 35 days at 25 °C
[45,46]”. This method does not require sample pretreatment. More information can be found in the
manufacturer´s web site at:
https://www.formulaction.com/en/products-and-technologies/technologies/static-multiple-lightscattering-s-mls (Static multiple light scattering)
https://www.formulaction.com/en/our-solutions/dispersibility-redispersion (video of S-MLS
technology and sample measurement)
Point 5: p-342: “the characteristics of the oil and emulsifier used…” (perhaps to specify which
one? e.g. concentration and type?)
Response 5: It is a general paragraph about effective aspects on emulsion particle size. The
characteristics of the oil and emulsifier used were specified in the second paragraph of the
Introduction section (e.g. type, viscosity, density, concentration, interfacial tension, etc.).
Point 6: References:
• A number of References (about 10%) are old ones of close or older than 20 years! (e.g. Nr
29, 35, 40, 59, 64, 65 etc.). Are they extremely important for the analysis/discussion or
perhaps could be replaced by most updated literature evidence?
• In References Nr 29, 67, please highlight date of publication in bold
Response 6: We thank the reviewer for this comment. According to this suggestion, References
[29], [35], [59], [60], [61], [64] and [65] have been removed and replaced in the revised manuscript
by most updated ones (e.g., References [29–32] and [63]). References have been renumbered
throughout the revised manuscript.
Furthermore, we highlighted date of publication in bold in Reference Nr 67 (65 in the revised
version), and also in Nr. 1 and 2.
Point 7: General comment: It could perhaps enhance quality if you include 1-2 images about the
optical characterisation of nanoemulsions e.g. indicating baseline/destabilisation etc
Response 7: Optical characterization of nanoemulsions is shown in Figure 4 (backscattering (BS)
profiles). The baseline is the dark blue line (first measurement for each emulsion, as indicated in the
time/color scale located at the right of each graph). Destabilization cause (i.e. creaming instability)
is also indicated in text (page 13, last paragraph).